Comparative transcriptome analyses of a late-maturing mandarin mutant and its original cultivar reveals gene expression profiling associated with citrus fruit maturation

Wang Lu 1 2
Hua Qingzhu 1
Ma Yuewen 1
Hu Guibing 1
Qin Yonghua qinyh@scau.edu.cn 1
1 State Key Laboratory for Conservation and Utilization of Subtropical Agro-bioresources, Key Laboratory of Biology and Genetic Improvement of Horticultural Crops-South China, Ministry of Agriculture, College of Horticulture, South China Agricultural University , Guangzhou , China
2 Yunnan Key Laboratory for Wild Plant Resources, Key Laboratory for Economic Plants and Biotechnology, Kunming Institute of Botany, Chinese Academy of Sciences , Kunming , China
Banada Padmapriya
Electronic publication date: 2017 May 18
Publication date: 2017
Volume: 5
Electronic Location ID: e3343
Received 2017 Jan 20; Accepted 2017 Apr 21
Copyright: ©2017 Wang et al.
Copyright year: 2017
Copyright holder: Wang et al.
License: This is an open access article distributed under the terms of the Creative Commons Attribution License, which permits unrestricted use, distribution, reproduction and adaptation in any medium and for any purpose provided that it is properly attributed. For attribution, the original author(s), title, publication source (PeerJ) and either DOI or URL of the article must be cited.
License URL: https://creativecommons.org/licenses/by/4.0/

Keywords: Citrus reticulata Blanco, Late maturity, RNA-Seq, Gene expression, NCED1

Funding: Science and Technology Planning Project of Guangdong Province 2010B020305007 Key Laboratory of Innovation and Utilization for Germplasm Resources in Horticultural Crops in Southern China of Guangdong Higher Education Institutes South China Agricultural University KBL11008 This work was supported by the Science and Technology Planning Project of Guangdong Province (2010B020305007), Key Laboratory of Innovation and Utilization for Germplasm Resources in Horticultural Crops in Southern China of Guangdong Higher Education Institutes, the South China Agricultural University (No. KBL11008). The funders had no role in study design, data collection and analysis, decision to publish, or preparation of the manuscript.

==============================
Characteristics of late maturity in fruit are good agronomic traits for extending the harvest period and marketing time. However, underlying molecular basis of the late-maturing mechanism in fruit is largely unknown. In this study, RNA sequencing (RNA-Seq) technology was used to identify differentially expressed genes (DEGs) related to late-maturing characteristics from a late-maturing mutant ‘Huawan Wuzishatangju’ (HWWZSTJ) (Citrus reticulata Blanco) and its original line ‘Wuzishatangju’ (WZSTJ). A total of approximately 17.0 Gb and 84.2 M paried-end reads were obtained. DEGs were significantly enriched in the pathway of photosynthesis, phenylpropanoid biosynthesis, carotenoid biosynthesis, chlorophyll and abscisic acid (ABA) metabolism. Thirteen candidate transcripts related to chlorophyll metabolism, carotenoid biosynthesis and ABA metabolism were analyzed using real-time quantitative PCR (qPCR) at all fruit maturing stages of HWWZSTJ and WZSTJ. Chlorophyllase (CLH) and divinyl reductase (DVR) from chlorophyll metabolism, phytoene synthase (PSY) and capsanthin/capsorubin synthase (CCS) from carotenoid biosynthesis, and abscisic acid 8′-hydroxylase (AB1) and 9-cis-epoxycarotenoid dioxygenase (NCED1) from ABA metabolism were cloned and analyzed. The expression pattern of NCED1 indicated its role in the late-maturing characteristics of HWWZSTJ. There were 270 consecutive bases missing in HWWZSTJ in comparison with full-length sequences of NCED1 cDNA from WZSTJ. Those results suggested that NCED1 might play an important role in the late maturity of HWWZSTJ. This study provides new information on complex process that results in the late maturity of Citrus fruit at the transcriptional level.

Introduction

Fruit maturity date is an important economic trait and selection of varieties with different harvest time would be advantageous to extend their storage period and market share. Citrus, one of the most important fruit crops, is a large-scale commercial production in the tropical and subtropical regions of the world. The total harvested area of citrus exceeds 8.8 million ha, with an annual yield of over 130 million tons in 2015 (Food and Agricultural Organization of the United Nations, 2014). Currently, harvest time for most citrus is mainly from November to December resulting in huge market pressure. Therefore, breeding of early- and late-maturing citrus varieties is essential to extend marketing season, meet the needs of consumers and ensure an optimal adaptation to climatic and geographic conditions.

Plant hormones play important roles in the regulation of fruit development and ripening (Kumar, Khurana & Sharma, 2014). Ethylene is known to be the major hormonal regulator in climacteric fruit ripening. In addition to ethylene, abscisic acid (ABA), auxin, gibberellin (GA) and brassinosteroid are involved in regulating fruit ripening. ABA plays an important role as an inducer along with ethylene signaling for the onset of fruit degreening and carotenoid biosynthesis during development and ripening process in climacteric and non-climacteric fruits (Leng et al., 2009; Sun et al., 2010; Jia et al., 2011; Romero, Lafuente & Rodrigo, 2012; Soto et al., 2013; Wang et al., 2016). ABA treatment can rapidly induce flavonol and anthocyanin accumulation in berry skins of the Cabernet Sauvignon grape suggesting that ABA could stimulate berry ripening and ripening-related gene expression (Koyama, Sadamatsu & Goto-Yamamoto, 2010). ABA also participates in the regulation of fruit development and ripening of tomato (Zhang, Yuan & Leng, 2009; Sun et al., 2011), cucumber (Wang et al., 2013), strawberry (Jia et al., 2011), bilberry (Karppinen et al., 2013), citrus (Zhang et al., 2014) and grape (Nicolas et al., 2014). Recent studies showed that ABA is a positive regulator of ripening and exogenous ABA application could effectively regulate citrus fruit maturation (Wang et al., 2016). Those results suggest that ABA metabolism plays a crucial role in the regulation of fruit development and ripening. In addition, fruit deterioration and post-harvest processes might influence fruit quality and ripening process. However, there are few reports involved in those processes. α-mannosidase (α-Man) and β-D-N-acetylhexosaminidase (β-Hex) are the two ripening-specific N-glycan processing enzymes that have proved that their transcripts increased with non-climacteric fruit ripening and softening (Ghosh et al., 2011). Genetic results have proved that 9-cis-epoxycarotenoid dioxygenase (NCED) is the key enzyme in ABA metabolism in plants (Liotenberg, North & Marion-Poll, 1999; Luchi et al., 2001). NCED1 could initiate ABA biosynthesis at the beginning of fruit ripening in both peach and grape fruits (Zhang et al., 2009). Silence of FaNCED1 (encoding a key ABA synthesis enzyme) in strawberry fruit could cause the ABA levels to decrease significantly and uncolored fruits and this phenomenon could be rescued by application of exogenous ABA (Jia et al., 2011). Suppression of the expression of SLNCED1 could result in the delay of fruit softening and maturation in tomato (Sun et al., 2012). Overexpression of ABA-response element binding factors (SlAREB1) in tomato could regulate organic acid and sugar contents during tomato fruit development. Higher levels of organic acid, sugar contents and related-gene expression were detected in SlAREB1-overexpressing lines in fruit pericarp of mature tomato (Bastías et al., 2011). However, there is little information available about the role of NCED1 genes in citrus fruit maturation (Zhang et al., 2014).

Bud mutant selection is the most common method for creating novel cultivars in Citrus. The ‘Huawan Wuzishatangju’ (HWWZSTJ) mandarin is an excellent cultivar derived from a bud sport of a seedless cultivar ‘Wuzishatangju’ (WZSTJ). Fruits of HWWZSTJ are mature in late January to early February of the following year, which is approximately 30 d later than WZSTJ (Qin et al., 2013; Qin et al., 2015). Therefore, the late-maturing mutant and its original cultivar are excellent materials to identify and describe the molecular mechanism involved in citrus fruit maturation. In this study, the highly efficient RNA-Seq technology was used to identify differentially expressed genes (DEGs) between the late-maturing mutant HWWZSTJ and its original line WZSTJ mandarins. DEGs involved in carotenoid biosynthesis, chlorophyll degradation and ABA metabolism were characterized. The present work could help to reveal the molecular mechanism of late-maturing characteristics of citrus fruit at the transcriptional level.

Materials & Methods

Plant materials

The late-maturing mutant ‘Huawan Wuzishatangju’ (HWWZSTJ) (Citrus reticulata Blanco) and its original cultivar ‘Wuzishatangju’ (WZSTJ) were planted in the same orchard in South China Agricultural University (23°09′38″N, 113°21′13″E). Ten six-year-old trees of each cultivar were used in this experiment. Peels (including albedo and flavedo fractions) from fifteen uniform-sized fresh fruits were collected on the 275th (color-break stage, i.e., peels turns from green to orange) and 320th (maturing stage) days after flowering (DAF) of HWWZSTJ and 275th (maturing stage) DAF of WZSTJ (Fig. S1) in 2012 and pools were named T3, T1 and T2, respectively. Peels from fifteen uniform-sized fresh fruits of HWWZSTJ and WZSTJ were collected on the 255th, 265th, 275th, 285th, 295th, 305th, 315th and 320th DAF in 2012 and used for expression analyses of candidate transcripts associated with chlorophyll, carotenoid biosynthesis and ABA metabolism. All samples were immediately frozen in liquid nitrogen and stored at −80 °C until use.

RNA extraction, library construction and RNA-Seq

Total RNA was extracted from peels according to the protocol of the RNAout kit (Tiandz, Beijing, China) and genomic DNA was removed by DNase I (TaKaRa, Dalian, China). RNA quality was analyzed by 1.0% agarose gel and its concentration was quantified by a NanoDrop ND1000 spectrophotometer (NanoDrop Technologies, Wilmington, DE, USA). RNA integrity number (RIN) values (>7.0) were assessed using an Agilent 2100 Bioanalyzer (Agilent Technologies, Santa Clara, CA, USA).

Construction of RNA-Seq libraries was performed by the Biomarker Biotechnology Corporation (Beijing, China). mRNA was enriched and purified with oligo (dT)-rich magnetic beads and then broken into short fragments. The cleaved RNA fragments were reversely transcribed to the first-strand cDNA using random hexamer primers. The second-strand cDNA was synthesized using RNase H and DNA polymerase I. The cDNA fragments were purified, end blunted, ‘A’ tailed, and adaptor ligated. The distribution sizes of the cDNA in the three libraries were monitored using an Agilent 2100 bioanalyzer. Finally, the three libraries were sequenced using an Illumina HiSeq™ 2500 platform.

Transcriptome assembly and annotation

Sequences obtained in this study were annotated in reference to the genome sequence of Citrus sinensis (Xu et al., 2013; Wang et al., 2014) using a TopHat program (Trapnell, Pachter & Salzberg, 2009). Functional annotation of the unigenes was performed using BLASTx (Altschul et al., 1997) and classified by Swiss-Prot (SWISS-PROT downloaded from European Bioinformatics Institute by Jan., 2013), Clusters of Orthologous Groups of Proteins Database (COG) (Tatusov et al., 2000), Kyoto Encyclopedia of Genes and Genomes Database (KEGG, release 58) (Kanehisa et al., 2004), non-redundant (nr) (Deng et al., 2006) and Gene Ontology (GO) (Harris et al., 2004). The number of mapped and filtered reads for each unigene was calculated and normalized giving the corresponding Reads Per Kilobases per Million reads (RPKM) values. DEGs between the two samples were determined according to a false discovery rate (FDR) threshold of <0.01, an absolute log2 fold change value of ≥1 and a P-value <0.01.

Gene validation and expression analysis

Data from RNA-Seq were validated using qPCR. All pigment-related (chlorophyll metabolism, carotenoid biosynthesis and ABA metabolism) uni-transcripts were selected to elucidate their expression patterns at all peel coloration stages of HWWZSTJ and WZSTJ with specific primers (Table S1). The citrus actin gene (accession No. GU911361.1) was used as an internal standard for the normalization of gene expression. Expression levels of all pigment-related uni-transcripts were determined using qPCR in an Applied Biosystems 7500 real-time PCR system (Applied Biosystems, CA, USA). A total of 20.0 µl reaction volume contained 10.0 µl THUNDERBIRD SYBR qPCR Mix (TOYOBO Co., Ltd.), 50×ROX Reference dye, 2.0 µl Primer Mix (5.0 µM), 6.0 µl ddH2O, and 2.0 µl cDNA (40 ng). The qPCR parameters were: 94 °C for 60 s then 40 cycles of 95 °C for 15 s, 55 °C for 15 s and 72 °C for 30 s. All experiments were performed three times with three biological replicates. Relative expression levels of selected transcripts were calculated by the 2−ΔΔCT method (Livak & Schmittgen, 2011).

All pigment-related genes (chlorophyll metabolism, carotenoid biosynthesis, ABA metabolism) were cloned using specific primers (Table S2). The 20.0 µl of reaction volume contained 2.0 µl 10×PCR buffer, 2.0 µl dNTP (2.0 mM), 0.2 µl of each primer (10 µM), 2.0 µl DNA (100 ng), 0.2 µl LA Taq and 13.4 µl ddH2O. PCR reaction procedure was 94 °C for 4 min then 35 cycles of 94 °C for 30 s, 55 °C for 30 s and 72 °C for 2 min, with a final 72 °C for 10 min. Nucleotide sequences of the pigment-related genes were analyzed using the National Center for Biotechnology Information (NCBI) Blast program (http://www.ncbi.nlm.nih.gov/BLAST). ORFs were made using the NCBI ORF Finder (http://www.ncbi.nlm.nih.gov/gorf/gorf.html). Alignments were done using ClustalX 1.83 and DNAMan software. Phylogenetic analysis of deduced amino acid sequences were performed using MEGA (version 5.0) and the Neighborjoining method with 1,000 bootstrap replicates.

Results

RNA-Seq analyses

To obtain differentially expressed genes (DEGs) between HWWZSTJ and WZSTJ, three libraries (T1, T2 and T3) were designed for RNA-Seq. As shown in Table 1, 26,403,257, 29,163,126, and 28,606,868 raw reads were obtained respectively from the three libraries. After removing low-quality bases and reads, a total of approximately 17.0 Gb clean reads were obtained. The GC contents for T1, T2 and T3 were 44.27%, 44.62% and 44.20%, respectively (Table 1). The range of most transcripts length was 100–200 bp (Fig. S2). Q30 percentage (percentage of sequences with sequencing error rate lower than 0.01%) for each sample was over 90% (Table 1).

Table 1 Summary of the sequencing data.

Samples	Total reads	Total base	GC content (%)	Q30 (%)	
T1	26,403,257	5,332,498,617	44.27	94.11	
T2	29,163,126	5,890,197,025	44.62	93.98	
T3	28,606,868	5,777,864,876	44.20	93.90	
Notes.

T1 HWWZSTJ (320 DAF)

T2 WZSTJ (275 DAF)

T3 HWWZSTJ (275 DAF)

A total of 44,664,047, 49,507,338 and 48,492,905 reads were mapped which accounted for 84.58%, 84.88% and 84.76% of the total reads, respectively (Table 2). Number of unique mapped reads accounted for 97.14% (T1), 97.25% (T2) and 97.19% (T3) of the total mapped reads compared with 2.86% (T1), 2.75% (T2) and 2.81% (T3) for multiple mapped reads, respectively. Those results suggested that the throughput and sequencing quality was high enough for further analyses.

Table 2 Summary of the transcriptome annotation compared with the reference genome of C. Sinensis (Xu et al., 2013).

Statistics libraries	T1	T2	T3	
	Number	Percentage	Number	Percentage	Number	Percentage	
Total reads	52,806,514	100.0%	58,326,252	100.0%	57,213,736	100.0%	
Mapped reads	44,664,047	84.58%	49,507,338	84.88%	48,492,905	84.76%	
Unique mapped reads	43,386,022	97.14%	48,146,871	97.25%	47,129,445	97.19%	
Multiple mapped reads	1,278,025	2.86%	1,360,467	2.75%	1,363,460	2.81%	
Pair mapped reads	39,251,294	87.88%	43,459,426	87.78%	42,663,447	87.98%	
Single mapped reads	4,574,673	10.24%	5,159,966	10.42%	4,969,429	10.25%	
Notes.

T1 HWWZSTJ (320 DAF)

T2 WZSTJ (275 DAF)

T3 HWWZSTJ (275 DAF)

Analyses of differentially expressed genes (DEGs)

DEGs were screened by comparison between any two of the three libraries using p < 0.01, FDR < 0.01 and Fold Change ≥ 2 as thresholds. A total of 2,687, 3,002 and 1,834 DEGs were obtained between the T1 and T3, T2 and T1, T2 and T3 libraries, respectively (Fig. 1A). Among those DEGs, 1,162, 1,567 and 770 were up-regulated and 1,525, 1,435 and 1,064 were down-regulated (Fig. 1B). Transcriptional levels of DEGs in HWWZSTJ on 320th DAF were lower than that on 275th DAF in HWWZSTJ suggesting that transcriptional levels of DEGs decreased during fruit maturation in HWWZSTJ (Fig. 1B).

Figure 1 Venn diagram (A) and histogram (B) of DEGs.

T1, HWWZSTJ (320 DAF); T2, WZSTJ (275DAF); T3, HWWZSTJ (275DAF).

Functional annotation of transcripts

A total of 299 new transcripts were annotated using five public databases (Nr, Swiss-Prot, KEGG, COG and GO). A summary of the annotations was shown in Table S3. Maximum number of annotation of differentially expressed transcripts (2,954) was in the Nr databases by comparison between T1 and T3, T2 and T1, T2 and T3, followed by GO databases (2,648) (Table S4). The differentially expressed transcripts were classified into three categories in GO assignments: cellular component, molecular function and biological process. DEGs between T1 and T3, T2 and T1, T2 and T3 were all significantly enriched in pigmentation, signaling and growth biological processes (Fig. S3A). Based on COG classifications, differentially expressed transcripts were divided into 25 different functional groups (Fig. S3B). DEGs between any two of the three libraries (T1-VS-T3, T2-VS-T1, T2-VS-T3) were assigned to 91, 100 and 91 KEGG pathways, respectively (File S1), and phenylalanine metabolism, porphyrin and chlorophyll metabolism, and flavonoid biosynthesis were the three significantly enriched biological processes (Table 3).

Table 3 Analyses of differentially expressed transcripts based on KEGG pathway.

#	Pathway	DEGs with pathway annotation (283)	All genes with pathway annotation (3516)	p_value	corr_p_ value	Pathway ID	
T1 vs T3	1	Phenylpropanoid biosynthesis	24 (8.48%)	82 (2.33%)	9.58e−09	8.71e−07	ko00940	
2	Photosynthesis	14 (4.95%)	46 (1.31%)	7.90e−06	7.19e−04	ko00195	
3	Plant-pathogen interaction	26 (9.19%)	130 (3.7%)	8.30e−06	7.56e−04	ko04626	
4	Plant hormone signal transduction	31 (10.95%)	180 (5.12%)	2.73e−05	2.49e−03	ko04075	
5	Phenylalanine metabolism	17 (6.01%)	72 (2.05%)	3.59e−05	3.26e−03	ko00360	
6	Photosynthesis-antenna proteins	7 (2.47%)	15 (0.43%)	7.45e−05	6.78e−03	ko00196	
7	Galactose metabolism	10 (3.53%)	45 (1.28%)	2.46e−03	2.23e−01	ko00052	
8	Starch and sucrose metabolism	21 (7.42%)	137 (3.9%)	2.66e−03	2.42e−01	ko00500	
9	Porphyrin and chlorophyll metabolism	8 (2.83%)	37 (1.05%)	7.83e−03	7.12e−01	ko00860	
10	Amino sugar and nucleotide sugar metabolism	14 (4.95%)	89 (2.53%)	1.06e−02	9.66e−01	ko00520	
T2 vs T1	1	Photosynthesis	18 (5.17%)	46 (1.31%)	1.1384e−07	1.1384e−05	ko00195	
2	Photosynthesis antenna proteins	10 (2.87%)	15 (0.43%)	1.5234e−07	1.5234e−05	ko00196	
3	Plant-pathogen interaction	32 (9.2%)	130 (3.7%)	5.5471e−07	5.5471e−05	ko04626	
4	Phenylpropanoid biosynthesis	22 (6.32%)	82 (2.33%)	7.9820e−06	7.9820e−04	ko00940	
5	Cyanoamino acid metabolism	9 (2.59%)	22 (0.63%)	1.2713e−04	1.2713e−02	ko00460	
6	Biosynthesis of unsaturated fatty acids	9 (2.59%)	29 (0.82%)	1.3656e−03	1.3656e−01	ko01040	
7	Phenylalanine metabolism	16 (4.6%)	72 (2.05%)	1.3850e−03	1.3850e−01	ko00360	
8	Flavonoid biosynthesis	10 (2.87%)	37 (1.05%)	2.3996e−03	2.3996e−01	ko00941	
9	Starch and sucrose metabolism	23 (6.61%)	137 (3.9%)	7.1760e−03	7.1760e−01	ko00500	
10	Stilbenoid, diarylheptanoid and gingerol biosynthesis	5 (1.44%)	14 (0.4%)	8.6944e−03	8.6944e−01	ko00945	
T2 vs T3	1	Photosynthesis	26 (10.48%)	46 (1.31%)	5.1027e−19	4.6434e−17	ko00195	
2	Photosynthesis-antenna proteins	11 (4.44%)	15 (0.43%)	1.8431e−10	1.6772e−08	ko00196	
3	Phenylpropanoid biosynthesis	24 (9.68%)	82 (2.33%)	6.3286e−10	5.7590e−08	ko00940	
4	Phenylalanine metabolism	16 (6.45%)	72 (2.05%)	2.6448e−05	2.4068e−03	ko00360	
5	Nitrogen metabolism	9 (3.63%)	32 (0.91%)	2.4845e−04	2.2609e−02	ko00910	
6	Flavone and flavonol biosynthesis	6 (2.42%)	15 (0.43%)	3.3744e−04	3.0707e−02	ko00944	
7	Cyanoamino acid metabolism	7 (2.82%)	22 (0.63%)	5.4299e−04	4.9412e−02	ko00460	
8	Stilbenoid, diarylheptanoid and gingerol biosynthesis	5 (2.02%)	14 (0.4%)	1.9788e−03	1.8007e−01	ko00945	
9	Glyoxylate and dicarboxylate metabolism	7 (2.82%)	28 (0.8%)	2.6144e−03	2.3791e−01	ko00630	
10	Flavonoid biosynthesis	8 (3.23%)	37 (1.05%)	3.5075e−03	3.1918e−01	ko00941	
11	Porphyrin and chlorophyll metabolism	8 (3.23%)	37 (1.05%)	3.5075e−03	3.1918e−01	ko00860	
12	Plant-pathogen interaction	18 (7.26%)	130 (3.7%)	3.9071e−03	3.5554e−01	ko04626	

Verification of the accuracy of the RNA-Seq data using qPCR

Twelve DEGs with significant differences from the three libraries were selected for verification of RNA-Seq data by qPCR. Linear regression analysis showed an overall correlation coefficient of 0.828, indicating a good correlation between qPCR results and the transcripts per kilobase million from the RNA-Seq data (Fig. S4).

Table 4 Analyses of transcripts involved in carotenoid biosynthesis, chlorophyll and ABA metabolism.

Gene ID	RPKM	Nr-annotation	
	T1	T2	T3		
Chlorophyll metabolism	
Cs3g03100	69.88	63.55	72.16	Probable glutamate-tRNA ligase [Arabidopsis thaliana]	
Cs8g01360	56.22	59.50	59.42	Glutamate-tRNA ligase 1 [Arabidopsis thaliana]	
Cs3g16730	78.03	66.88	70.76	Glutamyl-tRNA reductase 1 [Arabidopsis thaliana]	
Orange1.1t02623	89.64	138.40	164.18	Glutamate-1-semialdehyde 2,1-aminomutase 1, Chloroplastic [Arabidopsis thaliana]	
Cs3g19770	35.03	42.97	62.68	Delta-aminolevulinic acid dehydratase, chloroplastic [Arabidopsis thaliana]	
Cs7g13850	16.13	23.66	30.83	Porphobilinogen deaminase [Arabidopsis thaliana]	
Cs5g12440	21.59	19.34	27.86	Uroporphyrinogen decarboxylase 1, chloroplastic	
Cs7g30080	53.66	58.13	83.90	Uroporphyrinogen decarboxylase 2, chloroplastic	
Orange1.1t02279	37.48	71.77	87.12	Coproporphyrinogen-III oxidase, chloroplastic	
Cs5g06770	4.31	6.90	7.22	Oxygen-independent coproporphyrinogen-III oxidase 1	
Cs2g24910	21.98	25.25	31.77	Protoporphyrinogen oxidase, chloroplastic/mitochondrial	
Orange1.1t01782	35.37	56.42	53.75	Protoporphyrinogen oxidase, chloroplastic	
Cs9g13460	44.52	5.36	17.65	Magnesium-chelatase subunit H	
Cs2g30990	16.52	19.32	21.84	Magnesium-chelatase 67 kDa subunit	
Cs2g05100	118.46	82.35	183.85	Magnesium-chelatase subunit ChlI-1, chloroplastic	
Cs7g19710	6.74	3.42	18.65	Magnesium-protoporphyrin O-methyltransferase	
Cs6g16200	76.55	14.23	116.31	Magnesium-protoporphyrin IX monomethyl ester [oxidative] cyclase 1	
Cs1g06850	22.97	15.00	167.60	Protochlorophyllide reductase A, chloroplastic	
Cs5g16830	5.28	29.95	0.81	Chlorophyllase type 0	
Cs9g07520	30.05	13.07	20.43	Chlorophyllase type 0	
Cs6g08720	57.16	45.38	61.05	Bacteriochlorophyll synthase 34 kDa chain	
Cs3g19690	47.34	3.63	10.36	Chlorophyll synthase, putative [Ricinus communis]	
Cs4g15890	56.44	69.89	61.91	Chlorophyll (ide) b reductase NOL, chloroplastic	
Cs7g24010	5.24	11.18	13.54	Chlorophyll (ide) b reductase NOL, chloroplastic	
Cs8g15480	64.04	92.78	73.45	Pheophorbide a oxygenase, chloroplastic	
Cs1g22670	37.78	73.56	82.11	Red chlorophyll catabolite reductase, chloroplastic	
Carotenoid biosynthesis	
Cs6g15910	79.73	216.13	172.64	Phytoene synthase	
Orange1.1t02108	30.15	85.62	32.83	PREDICTED: phytoene synthase 2, chloroplastic-like [Vitis vinifera]	
Orange1.1t02361	50.61	63.09	64.41	Phytoene dehydrogenase, chloroplastic/chromoplastic	
Cs5g24730	47.11	59.11	58.21	15-cis-zeta-carotene isomerase, chloroplastic	
Cs3g11180	56.36	81.36	70.96	Phytoene dehydrogenase, chloroplastic/chromoplastic	
Cs6g13340	25.39	26.28	23.76	Prolycopene isomerase 1, chloroplastic	
Cs4g14850	7.43	5.56	10.07	Capsanthin/capsorubin synthase, chromoplast	
Orange1.1t00772	9.33	8.65	11.12	Capsanthin/capsorubin synthase, chromoplast	
Orange1.1t01058	38.85	39.93	39.95	Cytochrome P450 97B1, chloroplastic	
Cs9g19270	1038.75	1722.54	1359.83	Beta-carotene 3-hydroxylase 1, chloroplastic	
Cs1g22620	65.41	82.35	95.98	3-hydroxybenzoate 6-hydroxylase 1	
Orange1.1t04051	1.01	0.95	1.10	3-hydroxybenzoate 6-hydroxylase 1	
Cs5g26080	11.76	9.88	16.31	Violaxanthin de-epoxidase, chloroplastic	
Orange_new Gene_1755	0.45	3.82	9.60	Lycopene beta-cyclase [Citrus×paradisi]	
ABA metabolism	
Cs8g13780	4.47	3.20	4.51	Indole-3-acetaldehyde oxidase	
Cs6g01180	46.39	38.09	60.42	Xanthoxin dehydrogenase	
Cs2g03270	2.08	67.84	18.90	9-cis-epoxycarotenoid dioxygenase 2 [Citrus sinensis]	
Cs5g14370	28.65	20.09	8.45	Putative 9-cis-epoxycarotenoid dioxygenase 3 [Citrus sinensis]	
Cs7g14820	2.18	0.26	5.90	Carotenoid cleavage dioxygenase 4a [Citrus clementina]	
Cs9g11260	0.00	0.00	0.00	Carotenoid 9,10(9,10′)-cleavage dioxygenase 1	
Cs6g19380	60.57	30.48	9.86	ABA 8&apos; -hydroxylase [Citrus sinensis]	
Cs8g05940	2.05	3.80	3.20	Abscisic acid 8′-hydroxylase 1	
Cs8g18780	1.10	0.78	0.27	ABA 8&apos; -hydroxylase [Citrus sinensis]	

DEGs related to carotenoid biosynthesis, chlorophyll and ABA metabolism

Analyses of the expression data obtained through RNA-Seq revealed that most DEGs were involved in carotenoid biosynthesis, chlorophyll and ABA metabolism. The main transcripts involved in the three pathways were shown in Table 4 and heatmaps were made based on transcripts per kilobase million from the RNA-Seq data (Fig. 2). Three transcripts (Cs8g15480, Pheophorbide a oxygenase; Cs5g16830, Chlorophyllase type 0 and Cs3g19690, Chlorophyll synthase) involved in chlorophyll degradation, six transcripts (Cs3g19770, Delta-aminolevulinic acid dehydratase; Cs9g13460, Magnesium-chelatase subunit H; Cs2g05100, Magnesium-chelatase subunit ChlI-1; Cs7g19710, Magnesium-protoporphyrin O-methyltransferase; Cs6g16200, Magnesium-protoporphyrin IX monomethyl ester (oxidative) cyclase 1 and Cs1g06850, Protochlorophyllide reductase A) involved in chlorophyll biosynthesis, five transcripts (Cs6g15910, Phytoene synthase; Orange1.1t02108, phytoene synthase 2; Cs6g13340, Prolycopene isomerase 1; Cs4g14850/Orange1. 1t00772, Capsanthin/capsorubin synthase and Orange_new Gene_1755, Lycopene beta-cyclase) involved in carotenoid biosynthesis and four transcripts (Cs2g03270/Cs5g14370, 9-cis-epoxycarotenoid dioxygenase 2; Cs6g01180, Xanthoxin dehydrogenase; Cs7g14820, Carotenoid cleavage dioxygenase 4a and Cs6g19380, ABA 8&apos; -hydroxylase) involved in ABA metabolism were obtained (Fig. 2 and Table 4).

According to the result of expression and annotation analyses, thirteen transcripts i.e., BCHP, CRD1, CHLM, CHLH1, HEMFI/HEMF2, FC1, DVR, CAO, CLH, CCS, PSY, AB and NCED1 associated with chlorophyll metabolism, carotenoid biosynthesis and ABA metabolism were obtained with a Fold Change ≥2 and FDR <0.01 as screening standard (Table 5).

Figure 2 Heatmap of main transcripts from chlorophyll metabolism (A), chlorophyll synthesis (B), carotenoid biosynthesis (D) and ABA metabolism (C).

Expression analyses of candidate transcripts

Expression patterns of candidate transcripts associated with chlorophyll metabolism were analyzed between WZSTJ and HWWZSTJ at all fruit maturation stages (Fig. 3). Compared with WZSTJ, lower expression levels of ALAD1 and CLH were detected in HWWZSTJ at all fruit maturation stages. Expression of ALAD1 and CLH were increasing before fruit maturation and decreased thereafter in both WZSTJ and HWWZSTJ. The highest expression level of CLH was detected on the 295th DAF in HWWZSTJ, which was 20 d later than WZSTJ. Expression levels of CAO1 and PAO in HWWZSTJ was higher than that in WZSTJ. FC1 showed a decrease trend during fruit maturation of WZSTJ and HWWZSTJ. As for GluRS, HEMF1, HEMG and CHLM, they showed irregular expression patterns in WZSTJ and HWWZSTJ (Fig. 3).

Figure 3 Expression patterns of genes associated with chlorophyll metabolism in WZSTJ and HWWZSTJ at all fruit maturation stages.

(A) ALAD1; (B) CAO1; (C) CHLM; (D) CLH; (E) FC1; (F) GluRs; (G) HEMF1; (H) HEMG; (I) PAO.

Table 5 Analyses of DEGs associated with carotenoid biosynthesis, chlorophyll and ABA metabolism.

Gene ID	Symbols	
Chlorophyll metabolism	
Cs5g10740/ Cs2g26780	Geranyl acyl geranyl acyl diphosphate reductase (BCHP)	
Cs6g16200	Methyl magnesium protoporphyrin IX single cyclase (CRD1)	
Cs7g19710	Magnesium protoporphyrin IX methyl transferase (CHLM)	
Cs2g05100/ Cs9g13460	Mg-chelatase subunit D (CHLD)/Mg-chelatase subunit H (CHLH1)	
Orangel1.1t02279	Coproporphyrin oxidative decarboxylase (HEMFI/HEMF2)	
Cs4g18730	Ferrochelatase (FC1)	
Cs1g06850	Divinyl reductase (DVR)	
Cs3g19690	Chlorophyllide a oxygenase (CAO)	
Cs5g16830	Chlorophyllase (CLH)	
Carotenoid biosynthesis	
Orange_new Gene_1755	Capsanthin/capsorubin synthase (CCS)	
Orange1.1t02108/Cs6g15910	Phytoene synthase (PSY)	
ABA metabolism	
Cs3g23530	Abscisic acid 8′-hydroxylase (AB)	
Cs5g14370	9-cis-epoxycarotenoid dioxygenase (NCED1)	

Six carotenoid biosynthesis-related transcripts showed a trend from rise to decline at all fruit maturation stages of WZSTJ and HWWZSTJ (Fig. 4). The highest expression level of CCS was detected on the 295th DAF in HWWZSTJ, which was 20 d later than that of WZSTJ. Expression levels of PDS1, PSY3, PSY5, PSY6 and PSY7 in WZSTJ were higher than that of HWWZSTJ. PDS1 showed an increasing trend during fruit maturation of WZSTJ and HWWZSTJ and reached its maximum expression on the 295th DAF. PSY5 showed the highest expression levels on the 275th DAF compared to the highest expression levels of PSY3, PSY6 and PSY7 on the 265th DAF in both WZSTJ and HWWZSTJ. Expression levels of PSY5 were increasing before the 275th DAF and decreased thereafter. PSY3, PSY6 and PSY7 were up-regulation before the 265th DAF and decreased gradually thereafter (Fig. 4).

Figure 4 Expression patterns of genes associated with carotenoid biosynthesis in WZSTJ and HWWZSTJ at all fruit maturation stages.

(A) CCS; (B) PDS1; (C) PSY3; (D) PSY5; (E) PSY6; (F) PSY7.

Expression patterns of two candidate transcripts i.e., AB1 and NCED1 related to ABA metabolism were analyzed at all fruit maturation stages of WZSTJ and HWWZSTJ (Fig. 5). AB1 showed a trend from rise to decline during fruit maturation stages of WZSTJ and HWWZSTJ. The highest expression level of AB1 was obtained on the 295th DAF in HWWZSTJ, which was 20 d later than WZSTJ. Similar expression patterns of NCED1 were observed before the 295th DAF in HWWZSTJ and WZSTJ (Fig. 5). The expression level of NCED1 in HWWZSTJ was lower than that of WZSTJ during 275th DAF to 305th DAF. The highest expression level of NCED1 was detected on the 305th DAF of WZSTJ and significantly deceased thereafter (Fig. 5). However, the highest expression of NCED1 was at 295 DAF in HWWZSTJ. Results from expression analyses of candidate genes suggested that NCED1 might play a leading role in late-maturing characteristics of HWWZSTJ.

Figure 5 Expression patterns of transcripts associated with ABA metabolism in WZSTJ and HWWZSTJ at all fruit maturation stages.

(A) AB1; (B) NCED1.

Cloning and phylogenetic analyses of candidate genes

Full-length cDNA sequences of CLH and DVR from chlorophyll metabolism, PSY3, PSY5, PSY6, PSY7 and CCS from carotenoid biosynthesis, AB1 and NCED1 from ABA metabolism were cloned from HWWZSTJ and WZSTJ mandarins. There was one difference in base pair of CLH, PSY3 and PSY5 cDNA sequences between HWWZSTJ and WZSTJ (Figs. S5–S7). However, the amino acid sequences of CLH, PSY3 and PSY5 from HWWZSTJ was 100% identical to that from WZSTJ. There were 4, 6, 4, 3 and 17 bp difference between the sequences of DVR, CCS, PSY6, PSY7 and AB1 derived from HWWZSTJ and WZSTJ and this resulted in 2, 3, 3, 1 and 8 differences in the amino acids that would have been incorporated during translation of these transcripts (Figs. S8–S12). Compared with WZSTJ,there were 270 consecutive bases missing in cDNA sequence of the NCED1 from HWWZSTJ (Fig. 6). Phylogenetic analysis showed that CLH, DVR, PSY and NCED1 belonged to the same cluster, and their homology in comparison with similar sequences derived from other species is depicted in Figs. S13–S16. Results from sequence analyses suggested that deletion of 270 nucleotides in NCED1 maybe result in late-maturing characteristics of HWWZSTJ.

Figure 6 Alignments of cDNA (A) and amino acid (B) sequences of the NCED1 from HWWZSTJ and WZSTJ.

W, HWWZSTJ; P, WZSTJ

Discussion

Chlorophyll degradation, carotenoid biosynthesis and ABA metabolism play important roles in regulating citrus fruit maturation through a series of related genes or special signal network (Zhang et al., 2014). In this study, RNA-Seq technology was used to screen DEGs between a late-maturing mandarin mutant HWWZSTJ and its wild type WZSTJ during fruit maturation. DEGs between any two of the three libraries were significantly enriched in biological processes such as photosynthesis, phenylpropanoid biosynthesis, carotenoid biosynthesis, chlorophyll metabolism, ABA metabolism, starch and sucrose metabolism (Table 3). Thirteen maturing-related transcripts involved in carotenoid biosynthesis, chlorophyll degradation and ABA metabolism were selected for further analysis.

CLH is the key enzyme catalyzing the first step in the chlorophyll degradation. It can catalyze the hydrolysis of ester bond to yield chlorophyllide and phytol in the chlorophyll breakdown pathway (Jacob-Wilk et al., 1999; Tsuchiya & Takamiya, 1999). Jacob-Wilk et al. (1999) isolated a CLH encoding an active chlorophyllase enzyme and verified the role of CLH in chlorophyll dephytylation by in vitro recombinant enzyme assays. Expression level of CLH in Valencia orange peel was low and constitutive and did not significantly increase during fruit development and ripening (Jacob-Wilk et al., 1999). In the present study, a CLH was obtained from the transcriptome dataset. No difference was detected in the amino acid sequences of CLH between HWWZSTJ and WZSTJ. Expression levels of CLH were increasing prior to citrus fruit maturing, decreasing thereafter in both WZSTJ and HWWZSTJ. The highest expression level of CLH was detected on the 295th DAF in HWWZSTJ, which was 20 d later than that of WZSTJ (Fig. 3). Similar results were also observed in peels of the late-maturing mutant from Fengjie72-1 navel orange (Liu et al., 2006) and Tardivo clementine mandarin (Distefano et al., 2009). Those results suggested that CLH may balance between chlorophyll synthesis and its breakdown (Jacob-Wilk et al., 1999).

Citrus is a complex source of carotenoids with the largest number of carotenoids (Kato et al., 2004). Carotenoid contents and compositions are main factors that affect peel color of most citrus fruits (Tadeo et al., 2008). PSY is a regulatory enzyme in carotenoid biosynthesis (Welsch et al., 2000). PSY is present at low expression level in unripe (green) melon fruit, reaches its highest levels when the fruit turns from green to orange and persists at lower levels during later ripening stages (Karvouni et al., 1995). Liu et al. (2006) studied the mechanism underlying the difference between Fengwan (a late-maturing mutant) navel orange and its original cultivar (Fengjie72-1). The highest expression levels of some carotenoid biosynthetic enzymes in the peels of the late-maturing mutant occurred 30 d later than that of the original cultivar (Liu et al., 2006). In this work, PSY showed a trend from rise to decline at all fruit maturation stages of the late-maturing mutant HWWZSTJ and its original line WZSTJ. The expression levels of PSY3, PSY5, PSY6 and PSY7 in HWWZSTJ were lower than that in WZSTJ. These results demonstrated that the mutation in HWWZSTJ influenced the transcriptional activation of PSY.

ABA can be considered as a ripening regulator during fruit maturation and ripening. NCED, a key enzyme involved in ABA biosynthesis, plays an important role in fruit ripening of avocado (Persea americana) (Chernys & Zeevaart, 2007), orange (Citrus sinensis) (Rodrigo, Alquezar & Zacarías, 2006), tomato (Solanum lycopersicum) (Nitsch et al., 2009; Zhang, Yuan & Leng, 2009), grape (Vitis vinifera) and peach (Prunus persica) (Zhang et al., 2009). The NCED1 were expressed only at the onset stage of ripening in peach and grape, when ABA content became high (Zhang et al., 2009). Zhang et al. (2014) studied the mechanism of a spontaneous late-maturing mutant of ‘Jincheng’ sweet orange and its wild type through the comparative analysis. The highest expression of CsNCED1 was at 215 DAA in WT. In our study, expression levels of NCED1 increased prior to fruit maturing and decreased significantly thereafter in both HWWZSTJ and WZSTJ. The highest expression level of NCED1 was detected on the 305th DAF of WT (WZSTJ). Our results were consistent with previous findings that NCED1 plays the most important role in the ABA biosynthesis pathway during the fruit maturing process (Zhang et al., 2014). Deletion of nucleotides could cause a shift of the reading frame and truncated protein, which can result in natural mutants. Compared with the cDNA sequence of NCED1 from WZSTJ, there were 270 consecutive bases missing in HWWZSTJ (Fig. 6). Those results suggested that NCED1 might play an important role in late-maturing of HWWZSTJ. A high-efficient regeneration system for WZSTJ has been established (Wang et al., 2015) and further study on the role of NCED1 in citrus is being carried out through genetic engineering.

Conclusion

RNA-Seq technology was used to identify pigment-related genes from a late-maturing mandarin mutant HWWZSTJ and its original cultivar WZSTJ. Thirteen candidate transcripts related to chlorophyll metabolism, carotenoid biosynthesis and ABA metabolism were obtained. NCED1, a gene involved in ABA metabolism, is probably involved in the formation of late maturity of HWWZSTJ based on sequence and expression analyses. The present study opens up a new perspective to study the formation of late maturity in citrus fruit.

Supplemental Information

File S1 Supplementary Files

Click here for additional data file.

We thank Peng Li, Zixing Ye and Jietang Zhao for help and support with field management and technical assistance.

Additional Information and Declarations

Competing Interests

Author Contributions

Data Availability

The authors declare there are no competing interests.

Lu Wang performed the experiments, analyzed the data, contributed reagents/materials/analysis tools, wrote the paper, prepared figures and/or tables, reviewed drafts of the paper.

Qingzhu Hua and Yuewen Ma performed the experiments, contributed reagents/materials/analysis tools, reviewed drafts of the paper.

Guibing Hu analyzed the data, reviewed drafts of the paper.

Yonghua Qin conceived and designed the experiments, analyzed the data, wrote the paper, prepared figures and/or tables, reviewed drafts of the paper.

The following information was supplied regarding data availability:

The raw data has been supplied as File S1.

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
