# Peer review of "Comparative transcriptome analyses of a late-maturing mandarin mutant and its original cultivar reveals gene expression profiling associated with citrus fruit maturation"

_PeerJ, doi:10.7717/peerj.3343_

## Round 0.1 · original submission · Major Revisions

Dear Dr. Yongua Qin,

I agree at large with the reviewers and I recommend you to resubmit the article with your rebuttal.

I look forward to receive your revised manuscript.

Thank you

Best,

Priya Banada

Reviewer 1 ·

Basic reporting

The manuscript is generally all right. The manuscript is riddled with errors of syntax. I have tried addressing some of them and this is may be seen in the attached file.

There is also a paper Ya-Jian Zhang, Xing-Jian Wang, Ju-Xun Wu, Shan-Yan Chen,,
Hong Chen, Li-Jun Chai,, Hua-Lin Y (2014) Comparative Transcriptome Analyses
between a Spontaneous Late-Ripening Sweet Orange Mutant and Its Wild Type Suggest the Functions of ABA, Sucrose and JA during Citrus Fruit Ripening PLOS ONE which in effect deals with many of the things mentioned in this paper and much more

Experimental design

The experimental design is all right

Validity of the findings

The findings are all right . One would have wished more discussion on the role of the sequence differences in transcripts between the mutant and parent orange. It is not clear if the "bud mutation" results in both transcript levels and also activity levels (enzyme). I would also welcome more comparisons with the work of Zhang et al 2014 which is briefly mentioned towards the end of the manuscript.

The figure comparing the RNA seq data with qPCR needs elaboration as to what transcripts were chosen.

Additional comments

I would suggest extensive revisions in language. a common mistake is in not differentiating between observation and effect. I have tried to show how this may be carried out.

Placing your findings in a pathway chart would have made the paper more informative and easier to comprehend..

It would be better to express your RNA Seq data in “Transcripts Per Kilobase Million”

Annotated reviews are not available for download in order to protect the identity of reviewers who chose to remain anonymous.

Reviewer 2 ·

Basic reporting

.The manuscript may need some English editing. It is too long and may need to be shortened a bit with redundancies removed ( especially in discussion section).

References are sufficient and the background context is provided.However a little more information on fruit deterioration process and the enzymes involved post harvest would have been useful.

The data is sufficient as per the objective. However some images of fruits would have added to the validity of the data.

The manuscript is complete as per the title of the study. Again however, the focus has become too narrow in the manuscript and some details on processes occurring post harvest maybe required.

Experimental design

Yes

The design has not indcated whether the albedo and flavedo were separated.If not how did the albedo vary from the flavedo in the transcriptome?

Technical analysis is adequate. Explanation of the peak in figure 5 in the WZSTI variety and its implications are not clear.

Materials and methods adequate except as stated above.

Validity of the findings

The data is sufficient for this report.However absence of any protein data or enzyme activity makes the study a bit scientifically unsound.As is well know transcription is only a first step in the process of regulation of cellular pathways. Without at least some functional assays the relevance of the study cannot be established unambiguously.

Additional comments

Please include data on enzyme activity and protein profile for a few of the genes involved in the process.

Reviewer 3 ·

Basic reporting

- English language requires deep revision. Many gramatical errors found. Even in the first sentence of the abstract. Example 1: "Late ripening characteristics of a fruit is a good agronomic trait for extending both the harvest period and marketing time".

- Lack of references in the introduction to support statistics on the importance of citrus as fruit crop.

Experimental design

In Materials and Methods:
- Plant Material: is not clear if they used peels at the color-break stage for the WZSTJ wild-type. Only mentioned: breaker stage for the HWWZSTJ mutant, and ripening stages for both WZST and HWWZSTJ lines. Why was the breaker control not included in the RNAseq study?

- It seems that both RNAseq are unreplicated. Only 3 RNAseq libraries were performed using a pool of fruit. This experimental design is not acceptable for gene expression analyses, because the resulting data is not statistically supported. There has to be at least 3 biological replicates per treatment to obtain biologically-relevant data. Without these replicated data, the manuscript should not be published.

- What are the size of the reads? Was the sequencing performed for single-end of pair-end?

- How many biological replicated were used for the qPCR analyses?

- How did the authors prove that the actin gene is a stable reference gene throughout fruit development? did they test other reference genes for qPCR?

- What are the efficiencies of the primers used?

Validity of the findings

- I consider that this is a relevant study in the field, because provides a first characterization of biological processes that occur in a late-ripening mutant.

- However major deficiencies in the experimental design makes it difficult to assess the importance of the DEGs identified in this study. A more robust experimental system is and further validation are required. The qPCR section of the manuscript provides confirmation of several candidate transcripts but is not clear how these data correlates with the RNAseq results. A simple Pearson correlation and scatterplot may resolve some of these questions.

- The manuscript becomes really descriptive when the narrative relies on describing all the expression changes seen at different developmental points. A better connection between overall changes in gene expression patterns and biological processes is needed.

- The cloning and characterization of the NCED1 gene is the strongest section of the manuscript and should be given more importance in the manuscript. The 270 bp deletion seems to be the major finding of the study.

- Authors should address in the discussion the fact that they are using the "orange" genome instead of the "mandarin" genome, therefore they might be loosing several genes that are not present in the mapping genome.

Additional comments

Authors should be careful in using the terms: "maturity" and "ripening". They refer to different developmental processes in fruit and are not interchangeable.

---

## Round 0.2 · accepted · Accept

Dear Dr. Qin,

Thank you for your resubmission. The manuscript is substantially improved and thanks for addressing the reviewers concerns. Congratulations! I am happy to recommend your article for publication in PeerJ.